# The Effect of Diosmin, Escin, and Bromelain on Human Endothelial Cells Derived from the Umbilical Vein and the Varicose Vein—A Preliminary Study

**DOI:** 10.3390/biomedicines11061702

**Published:** 2023-06-13

**Authors:** Lukasz Gwozdzinski, Joanna Bernasinska-Slomczewska, Pawel Hikisz, Anna Wiktorowska-Owczarek, Edward Kowalczyk, Anna Pieniazek

**Affiliations:** 1Department of Pharmacology and Toxicology, Medical University of Lodz, 90-752 Lodz, Poland; anna.wiktorowska-owczarek@umed.lodz.pl (A.W.-O.); edward.kowalczyk@umed.lodz.pl (E.K.); 2Department of Oncobiology and Epigenetics, Faculty of Biology and Environmental Protection, University of Lodz, 90-236 Lodz, Poland; joanna.bernasinska@biol.uni.lodz.pl (J.B.-S.); pawel.hikisz@biol.uni.lodz.pl (P.H.); anna.pieniazek@biol.uni.lodz.pl (A.P.)

**Keywords:** varicose vein, human endothelial cells, diosmin, escin, bromelain

## Abstract

In this study, we investigated the properties of human varicose vein (VV) endothelial cells (HVVEC) in comparison to the human umbilical vein endothelial cells (HUVEC). The cells were treated with three bioactive compounds with proven beneficial effects in the therapy of patients with VV, diosmin, escin, and bromelain. Two concentrations of tested drugs were used (1, 10 mg/mL), which did not affect the viability of either cell type. Escin led to a slight generation of reactive oxygen species in HUVEC cells. We observed a slight release of superoxide in HVVEC cells upon treatment with diosmin and escin. Diosmin and bromelain showed a tendency to release nitric oxide in HUVEC. Using membrane fluorescent probes, we demonstrated a reduced fluidity of HVVEC, which may lead to their increased adhesion, and, consequently, a much more frequent occurrence of venous thrombosis. For the first time, we show the mechanism of action of drugs used in VV therapy on endothelial cells derived from a VV. Studies with HVVEC have shown that tested drugs may lead to a reduction in the adhesive properties of these cells, and thus to a lower risk of thrombosis.

## 1. Introduction

Varicose veins (VV) are swollen, twisted blood vessels that appear under the surface of the skin in the form of blue and purple bulges, most often in the lower limbs. Varicose veins are the result of blood stagnation, or slow or turbulent venous blood flow in the vessels when their walls are weak and the valves are not working properly. They cause swelling of the lower limbs, hyperpigmentation of the skin, and, in more severe cases, chronic venous ulcers and a high risk of venous thrombosis. Some of the symptoms associated with varicose veins can be relieved with medication, sclerotherapy, laser therapy, or surgery.

Occasionally, in cases of severe varicose veins, blood clot formation may become a serious health problem, leading to venous thrombosis [1]. Varicose veins are common and occur in approximately 35% of US adults in the general population [2]. In turn, in the United Kingdom, it is estimated that patients with VV constitute 40% of the total population [3].

Varicose veins are the result of many factors, such as the age of patients, aging processes, overweight, poor physical condition, and a standing lifestyle. Varicose veins are changed, weakened veins and are the symptom of chronic venous insufficiency. As hydrostatic blood pressure increases, the vein becomes enlarged and the valves that keep the blood moving in one direction stop working properly, leading to stagnation and hypoxia. Under low oxygen conditions, many factors are expressed, such as hypoxia inducible factors (HIF-1, HIF-2), but also matrix metalloproteinase (MMP) inducers or activators, for example, the matrix metalloproteinase inducer. The activities of the matrix metalloproteinases cause inflammation and damage to endothelial cells covering the lumen of the vessel, and the appearance of structural and functional changes in the vein wall [4]. The influence of MMPs on the degradation of the extracellular matrix has also been demonstrated, which in turn causes a significant remodeling of the venous tissue, leading to structural and degenerative changes in the vein wall and valve dysfunction. The MMPs can also induce early changes in endothelial and venous smooth muscle function and venous dilation. In turn, leukocyte infiltration and vein wall inflammation further increase the number of MMPs, vein wall dilation, valve degradation, and the further development of chronic venous disease (CVD), including varicose veins [5].

Studies carried out on cell cultures and ex vivo explants have shown a hypoxia-induced activation of leukocytes and endothelial cells, which secrete mediators involved in the remodeling of the vein wall. Similar mediators are released in the case of varicose veins [6]. Under hypoxia, venous endothelial cells release greater amounts of HIF, which regulate many genes related to oxygen homeostasis. In hypoxia, vascular endothelial growth factor (VEGF) and endothelial isoforms of nitric oxide synthase (eNOS) are expressed, and prostaglandin I2 and cyclooxygenase-2 (COX-2) are increased. It was also found that polymorphonuclear nuclear (PMN) cells in hypoxia generate large amounts of superoxide and leukotriene B 4 [6]. Under hypoxic conditions, an increase in xanthine dehydrogenase (XD) and xanthine oxidase (XO) activity was observed in cultured pulmonary arterial endothelial (EC) cells without a change in the XD/XO ratio. Hypoxia can lead to oxidative and inflammatory damage to the endothelium [7]. It has been shown that reactive oxygen species (ROS) play an important role in the pathogenesis of a varicose vein. ROS are released by activated leukocytes (mainly neutrophils), which adhere to endothelial cells on the vessel wall [8,9]. In patients with a VV, a disturbed antioxidant defense mechanism in the blood, a decreased activity of superoxide dismutase (SOD) and total antioxidant status in plasma, and the malondialdehyde (MDA) release of an oxidative stress marker have been observed [9,10]. A decreased activity of SOD and the total antioxidant status and an increased MDA level were observed in the vein wall in a varicose vein in comparison to a normal vein [8]. On the other hand, there was an increase in antioxidant defense in the wall of varicose veins and an increase in the SOD activity [9]. In addition, ROS led to damage to the subendothelial tissue, smooth muscle cell hyperplasia, and an increase in endothelial permeability [10].

Bromelain is an extract obtained from the pineapple fruit (*Ananas comosus*), which includes enzymes such as thiol endopeptidases and phosphatases, glucosidases, peroxidase, cellulase, and other components such as glycoprotein and protease inhibitors [11,12]. Bromelain has antimicrobial, anticoagulant, antitumor, and anti-inflammatory properties [11,12]. This drug can be used in the treatment of many inflammatory diseases, cardiovascular diseases, coagulation and fibrinolysis disorders, infectious diseases, neoplastic diseases, and in oral surgery [13]. Bromelain is effective in reducing the development of inflammation and swelling. It was recently shown that bromelain can be effectively used in patients undergoing alveolar preservation surgery after tooth extraction [14]. However, the molecular mechanism of bromelain remains unclear.

Escin is a mixture of saponin found in horse chestnut seeds (*Aesculus hippocastanum* L., *Sapindaceae*) containing cryptoescin, α-escin, and β-escin isomers that have antioxidant properties [15]. It exerts an anti-inflammatory effect, inhibiting the formation of edema, and it restores the elasticity and proper tension of blood vessels, improving blood flow. Escin is used in the treatment of chronic venous insufficiency of the lower extremities and thrombophlebitis in varicose veins or hemorrhoids [16].

Diosmin belongs to a group of glycosylated flavonoids and was first isolated from the plants of the Rutaceae family. It is composed of diosmetin (aglycone) and a sugar part, which is the disaccharide 6-O-α-L-rhamnosyl-D-glucose. This flavonoid, as a phlebotonic drug, is used to improve the condition of the veins with its anti-inflammatory and antioxidant properties. It is used in chronic venous insufficiency and in varicose veins to slow the progression of the disease. It improves venous tone, reduces the inflammatory response and capillary leakage, and improves microcirculation [17].

This paper compares the effects of plant-derived compounds commonly used in venous diseases, such as diosmin, escin, and bromelain, on the endothelial cells derived from umbilical vein and varicose vein patients (HUVEC and HVVEC cells). The obtained endothelial cells were tested with diosmin, escin, and bromelain for viability, various reactive oxygen species generation, and changes in cell membrane fluidity.

## 2. Materials and Methods

### 2.1. Chemicals

Diosmin (3′,5,7-Trihydroxy-4′-methoxyflavone 7-rutinoside), escin (β-D-Glucopyranosyl-(1→2)-[β-D-glucopyranosyl-(1→4)]-(22α-(acetyloxy)-16α,24,28-trihydroxy-21β-{[(2*Z*)-2-methylbut-2-enoyl]oxy}olean-12-en-3β-yl β-D-glucopyranosiduronic acid), and bromelain from a pineapple stem were purchased from Sigma-Aldrich (St. Louis, MO, USA). All the investigated compounds were dissolved according to the suggestions of the manufacturers. Diosmin was dissolved in dimethyl sulfoxide (DMSO); the final concentration of the solvent at the highest concentration did not exceed 0.02%. Escin was dissolved in methanol, and similarly the final concentration of the solvent at its highest did not exceed 0.02%. An additional control for solvents at 0.02% concentration was performed. Bromelain was dissolved in PBS. Resazurin, 1-(4-(trimethylamino)phenyl)-6-phenylhexa-1,3,5-triene (TMA-DPH), 11-(Dansylamino)undecanoic acid (DAUDA), 2′,7′-dichlorofluorescin diacetate (H_2_DCFDA), dihydroethidium (DHEt), and diaminofluorescein-FM (DAF-FM) were obtained from Sigma-Aldrich (St. Louis, MO, USA). Sodium salts and other reagents were of the highest available purity and were purchased from POCH (Gliwice, Poland).

### 2.2. Cell Culture

The experiments were conducted on the two types of endothelial cells: human umbilical vein endothelial cells (HUVEC) and human varicose vein endothelial cells (HVVEC). HUVEC cells were isolated from umbilical cord veins obtained from six healthy volunteers directly after labor. The varicose vein endothelial cells were isolated from varicose veins obtained during surgery from six patients suffering from a chronic venous disease (sex (M/F 4/2), age (54.4 ± 10.6 years), body mass index (BMI) (27.7 ± 3.0 kg/m^2^)). The obtained varicose veins were categorized as C_2_E_s_A_3_P_r_ class using the CEAP classification. None of these patients used any phlebotropic drugs or any drugs that could affect the blood coagulation pathways.

Both types of the endothelial cells were isolated by collagenase type II (Gibco, Life technologies, Merelbeke, Belgium) digestion, according to Jaffe’s protocol [18], and used for the experiments at passage 3–4. The cells were cultured in an MCDB131 medium (Corning B.V. Life Science, Amsterdam, Netherlands) containing 10 ng/mL of the epidermal growth factor (Merck Millipore, Darmstadt, Germany), 2 mM glutamine (Corning B.V. Life Science, Amsterdam, Netherlands), 10% heat-inactivated fetal bovine serum (Biowest SAS, Nuaillé, France), and antibiotics (penicillin/streptomycin) (Biowest SAS, Nuaillé, France).

For comparative purposes, Figure 1 presents the microscopic images of the cultured HUVEC and HVVEC cells.

The culture flasks were incubated at 37 °C in a 95% air/5% CO_2_ atmosphere saturated with H_2_O. The cells were fed twice a week with a complete change of a fresh culture medium. For subculture, the cells were harvested with 0.25% trypsin–ethylenediaminetetraacetic acid sodium salt (EDTA) solution (Biowest SAS, Nuaillé, France). The study was conducted in accordance with the rules of the Declaration of Helsinki, and it conformed to the ethical principles set out by the Belmont Report, Ethical Principles and Guidelines for the Protection of Human Subjects of Research. The approval for the investigation was obtained from the Bioethical Committee of The Medical University of Lodz. All participants signed an informed consent form prior to enrolment in the study.

### 2.3. The Cytotoxicity Test

The cytotoxicity of the investigated compounds was estimated spectrophotometrically, using a resazurin reduction assay. Resazurin is a blue dye which is reduced by living cells to the pink colored resorufin by cytosolic, mitochondrial, and microsomal redox enzymes. Dye reduction is highly correlated with the number of viable and metabolically active cells [19]. The cells were seeded into 96-well plates at a density of 10 × 10^3^ in a 100 μL culture medium per well, 24 h before the experiment, and then they were exposed for 24 h to a range of concentrations from 1.25 μg/mL to 50 μg/mL. After the treatment, the cells were washed with phosphate-buffered saline (PBS), then the resazurin solution was added at a final concentration of 12.5 μg/mL, and the plate was returned to the incubator for another 2 h. The absorbances of resazurin and resorufin were measured, respectively, at 570 and 600 nm with a BioTek microplate reader.

### 2.4. Cell Treatment Conditions

The cells were seeded into Petri dishes at a suitable density for the performed assay and incubated for 24 h with diosmin, escin, or bromelain at two concentrations: 1 μg/mL and 10 μg/mL. The control cells were treated with a corresponding volume of PBS, methanol, or DMSO (instead of the investigated compounds). After the incubation, the cells were washed with PBS and used for the following measurements.

### 2.5. Measurement of Reactive Oxygen and Nitrogen Species

The measurement of the intracellular ROS and reactive nitrogen species (RNS) production in the endothelial cells was recorded by changes in the fluorescence of three different fluorescent probes. The H_2_DCFDA and DHEt are fluorescent probes, which may react with several reactive oxygen species. The cell-permeant H_2_DCFDA passively diffuses into the cells, and is retained at the intracellular level after cleavage by intracellular esterases. Upon oxidation by ROS (including hydrogen peroxide, hydroxyl radicals, and nitrogen dioxide radicals, but also hypochlorous acid and peroxynitrite), the nonfluorescent H_2_DCFDA is converted to the highly green fluorescent product 2′,7′-dichlorofluorescein (DCF) (absorption/emission: 504/529 nm) [20]. DHEt also possesses the ability to freely permeate the cell membranes, and is mainly used to monitor the superoxide production. Dihydroethidium itself shows a blue fluorescence (absorption/emission: 370/420 nm), but, in the cell cytoplasm, it is oxidized by superoxide to 2-hydroxyethidium, which becomes red fluorescent (absorption/emission: 535/610 nm) upon DNA intercalation [21]. The DAF-FM is a cell permeant reagent commonly used to detect and quantify even low concentrations of reactive nitrogen species, especially nitric oxide (^•^NO). DAF-FM is essentially nonfluorescent until it reacts with ^•^NO to form a fluorescent benzotriazole (absorption/emission: 495/515 nm) [22]. The final concentration of the fluorescent labels in the solution was 5 μM. The fluorescence intensity was measured with a Cary Eclipse fluorescence spectrophotometer (Varian, Inc., Cary Eclipse, Fluorescence Spectrometer, USA), after 30 min of incubation under culture conditions, at the following excitation/emission wavelengths: 504/529 nm for H_2_DCFDA, 535/610 nm for DHEt, and 495/515 nm for DAF-FM. The fluorescence intensity was normalized to protein concentrations and expressed as a percentage of the control.

### 2.6. Measurement of Membrane Fluidity

The membrane fluidity of the endothelial cells was monitored by changes in the fluorescence anisotropy of two fluorescent membrane probes, TMA-DPH and DAUDA [23]. The location of the fluorescent labels in the specific regions of the cell membrane allows the evaluation of the changes in the fluidity, and the identification of the region in which these changes occur. DAUDA was located in the hydrophobic region of the lipid membrane, while the TMA-DPH probe was near the hydrophilic part of the lipid bilayer. An increase in the fluorescence anisotropy of the label is a consequence of a decrease in the membrane fluidity [24]. The final concentration of the fluorescent labels in the solution was 1 μM. The probe mobilities in the lipid membrane were analyzed after 10 min of incubation under culture conditions at the following excitation/emission wavelengths: 355/516 nm for DAUDA, and 355/430 nm for TMA-DPH. Fluorescence anisotropy was measured with a Cary Eclipse fluorescence spectrophotometer (Varian, Inc.).

### 2.7. Statistical Analysis

The results represent mean ± standard deviation (SD). The normality of the data was analyzed using the Kolmogorov–Smirnov test, and the homogeneity of variance was tested with a Brown–Forsyth test. The significance of the differences between the groups was estimated with a one-way ANOVA and a post hoc Tukey test. The statistical analysis was performed using Statistica v. 13.1 (StatSoft Polska, Krakow, Poland).

## 3. Results

### 3.1. Cytotoxicity of the Investigated Compounds

The cytotoxicity of bromelain, escin, and diosmin to the endothelial cells in the concentration range from 1.25 µg/mL to 50 µg/mL using resazurin was determined. Reduced cell survival was observed after 24 h of cell incubation with diosmin and escin; however, the cytotoxicity of escin was greater. (Figure 2A,B). In addition, a significant reduction in the number of living varicose vein endothelial cells was observed after treatment with escin and diosmin at concentrations ≥ 15 μg/mL. Diosmin had a slightly greater effect on HUVEC viability than on HVVEC, reducing the cell survival by 15% and 12%, respectively (Figure 2A). Meanwhile, escin concentrations of 35 μg/mL and 50 μg/mL decreased HUVEC and HVVEC viability by approx. 22% and 30% (Figure 2B).

In case of bromelain, its lower concentration did not affect the human umbilical vein endothelial cell or the vascular endothelial cell survival. The highest concentrations (35 μg/mL and 50 μg/mL) of bromelain reduced HUVEC and HVVEC viability by approx. 8–10%. However, these data were not statistically significant (Figure 2C). Thus, in further experiments, endothelial cells were treated with diosmin, escin, or bromelain at two concentrations, 1 μg/mL and 10 μg/mL, which did not exhibit any influence on the viability of the two cell types. Additionally, we observed a lack of cytotoxicity in all the solutions used to dissolve the test compounds.

### 3.2. Reactive Oxygen and Nitrogen Species

In this work, we also investigated the effect of the plant-derived drugs on the generation of reactive oxygen species in the two types of cells using three fluorescent probes. We measured the total amount of ROS using the H_2_DCFDA probe. It was shown that H_2_DCFDA reacted with superoxide (O_2_^•−^), hydrogen peroxide (H_2_O_2_), hypochlorous acid (HClO), peroxynitrite (ONOO^−^), and nitrogen dioxide (^•^NO_2_) [25]. For the detection of O_2_^•−^, the selective DHEt probe was used, while the nitric oxide was identified using the DAF-FM probe. Potentially, ROS could have been released after HUVECs. The varicose vein endothelial cells were incubated with diosmin, escin, and bromelain, and we measured their presence in our study. Generally, it is accepted that diosmin, escin, and bromelain have antioxidant properties. However, we checked the pro-oxidizing nature of these drugs on both types of cells. The cells were exposed to bromelain, diosmin, or escin for 24 h at two concentrations, 1 µg/mL and 10 µg/mL, and then the level of ROS was measured. In the case of the HUVEC treated with escin, a slight increase in the fluorescence of H_2_DCFDA was observed (Figure 3). A slight increase in the fluorescence of this probe was also found for HVVEC treated with bromelain. However, these results were statistically insignificant.

Another fluorescent probe, DHEt, showed a slight increase in the generation of superoxide anion radicals in HVVEC after diosmin and escin treatment, but these results were statistically insignificant (Figure 4).

An insignificant increase in DAF-FM fluorescence in HVVEC treated with 10 µg/mL of escin was observed (Figure 5). An increase in the fluorescence of this probe was also found in both types of cells after the treatment with bromelain. We observed the release of ^•^NO in HVVEC cells after the treatments with diosmin and bromelain. These results are indicative of the release of nitric oxide in these cells. However, these results were not statistically significant.

All the obtained results showed that the tested compounds caused a slight increase in the reactive oxygen species in human umbilical vein endothelial cells and varicose vein endothelial cells. No significant changes in the level of ROS were observed for the three compounds and concentrations used. The expected lack of change in the increased level of reactive oxygen/nitrogen species in cells incubated with test compounds is a confirmation of our earlier studies, where it was shown that the tested compounds did not present cytotoxic properties toward the endothelial cells of varicose veins and umbilical veins in concentrations up to 10 µg/mL.

### 3.3. Cell Membrane Fluidity

Changes in the plasma membrane fluidity of human umbilical vein endothelial cells (HUVEC) and varicose vein endothelial cells treated with diosmin, escin, and bromelain were investigated using two fluorescent probes located at different depths of the membrane lipid monolayer: TMA-DPH is located in the surface of the lipid membrane, and DAUDA is located in the hydrophobic core [23].

Comparing the fluorescence anisotropy (r) values for both probes, we showed the differences in the membrane fluidity of HUVECs and HVVECs. For the TMA-DPH probe and HUVECs the r was 0.280 ± 0.014, and for HVVECs the r was 0.297 ± 0.03 (Figure 6). Figure 6 shows an upward trend in the anisotropy for the HUVEC cells treated with diosmin, while the trend for the HVVEC cells was exactly the opposite. Similar results were obtained for escin and bromelain; however, these results were statistically insignificant.

The influence of the tested compounds on the membrane fluidity was also studied using the DAUDA probe located in a deeper region of the lipid bilayer. The differences in the fluorescence anisotropy between HUVEC and HVVEC in the plasma membrane were greater for the DAUDA probe, 0.278 ± 0.049 and 0.320 ± 0.030, respectively (Figure 7). These results show that HVVEC cells’ plasma membranes are stiffer than those of HUVEC, and suggest that the investigated compounds can modulate the fluidity of plasma membranes in both cell types. In this case, the results were also statistically insignificant.

The obtained results indicate that the tested compounds did not cause statistically significant changes in the endothelial cell membrane fluidity. No changes were observed for either of the concentrations (1 and 10 µg/mL). In addition, no change in the fluidity of the membrane of endothelial cells exposed to the tested compounds was observed for both the hydrophilic and the hydrophobic regions of the lipid monolayer.

## 4. Discussion

Chronic venous disease (CVD) is associated with pathological changes in the vein wall as well as with the dysfunction of vein valves. This disease is characterized by the appearance of varicose veins, the hyperpigmentation of lower limb skin, venous leg ulcers, and skin lesions. The primary stage of CVD is thought to be faulty venous valves causing blood stagnation leading to increased and sustained venous hypertension [26,27]. Vein morphology studies have shown that valve damage is a consequence of the dilatation of the venous wall and the shortening of the valve leaflet, leading to incomplete valve closure and subsequent reflux. In addition, valvular studies showed an increased leukocyte infiltration by granulocytes, monocytes, and T cells. The expression of P-selectin and ICAM-1, two adhesion molecules on the endothelial cell membrane, was also observed on the endothelial cells of the saphenous vein wall [28].

Slow blood flow, or blood stagnation in varicose veins, leads to hypoxia and inflammation within the vessel wall. Blood samples collected from varicose veins showed an increased concentration of interleukin-6, fibrinogen, and hemoglobin in comparison with blood samples from the antecubital vein of the same patient [29]. Under these conditions, purine metabolism begins with ATP and ends with hypoxanthine and xanthine. However, during the conversion of hypoxanthine to xanthine with the participation of xanthine oxidase, the superoxide molecule is released. The second O_2_^•−^ molecule is released during the oxidation of xanthine to uric acid [30]. Both oxidative stress and inflammation exist in a varicose vein. It has been reported that hypoxia and cytokines, such as TNF-α, IFN-γ, IL-6, and IL-1, may activate xanthine oxidoreductase (XOR) gene transcription [31,32]. At the same time, the adhesion and activation of neutrophils and monocytes, which additionally release ROS, take place [33,34]. ROS can lead to endothelial dysfunction and the development of atherosclerosis, inflammation, chronic venous insufficiency, and also thrombosis. It has also been shown that endothelial dysfunction plays a crucial role in varicose veins, and deep venous thrombosis (DVT) [35].

Unstimulated endothelial cells express intercellular adhesion molecules (ICAM). In contrast, E-selectin and vascular cell adhesion molecules (VCAM) are present only on activated endothelial cells. In inflammation, the cytokines (TNF-α, IL1-β and IFN) acting on endothelial cells can induce the synthesis and expression of E-selectin and VCAM, as well as an increase in the ICAM levels [36]. Interactions between selectins and their respective ligands are involved in the adhesion of leukocytes to the EC under flow conditions [37].

Varicose veins are associated with elevated levels of inflammatory and prothrombotic markers [38,39]. Inflammation is believed to be associated with the pathophysiology of DVT (deep vein thrombosis), PE (pulmonary embolism), and PAD (peripheral artery disease) [40]. The inflammation in a varicose vein can also damage the veins, causing blood clots to form. Adult patients with varicose veins are at a significantly increased risk of developing a DVT [41]. It was found that the interactions between blood cells and the vessel wall belong to the main hematological factors determining hemostasis. Plasma factors, red blood cells, leukocytes, and endothelial cells are involved in thrombus formation. In inflammation, leukocytes play an important role in endothelial dysfunction, and, consequently, in the changes to the vessel wall, and the increase in coagulation potential through the production of the tissue factor [42].

### 4.1. Endothelial Cells Viability

We showed that diosmin and escin concentrations up to 10 µg/mL did not decrease the viability of the two investigated types of cells. In contrast, higher concentrations of diosmin (>35 µg/mL) led to a significant decrease in the viability of both types of cells, but a higher cytotoxicity was observed for HUVEC than for HVVEC cells. On the other hand, for escin, a greater decrease in cell viability was found at the concentration of 25 µg/mL. Escin showed higher cytotoxicity to HVVEC than HUVEC cells. Moreover, bromelain showed the lowest cytotoxic effect compared to diosmin and escin.

It has been shown that diosmin supports microcirculation, increases venous tone, venous elasticity, and capillary resistance, and reduces capillary filtration and capillary hyperpermeability, and has a beneficial effect on lymphatic drainage [43,44]. Moreover, it also has an anti-inflammatory effect and alleviates oxidative stress, and is considered an important drug in the treatment of chronic venous disorders (CVD) [45]. Diosmin is a drug used to treat venous diseases, such as chronic venous insufficiency, varicose veins, spider veins, venous ulcers, and leg edema. However, the mechanism of action of diosmin is not defined. In addition to its vasoprotective effects, diosmin has the ability to influence the production of nitric oxide by the endothelium [46,47]. The antioxidant properties of diosmin, escin, and bromelain, as a factor alleviating oxidative stress initiated by various chemical agents or/and ionizing radiation, have been demonstrated in many studies, both in vivo and in vitro [48,49,50,51,52]. The administration of diosmin to patients suffering from CVD has led to a significant reduction in the level of TNF-alpha, vascular endothelial growth factors (VEGF-A and VEGF-C), angiostatin, interleukin 6 (IL-6), and fibroblast growth factor 2 (FGF2), and an increase in plasma angiostatin levels after three months of treatment [53]. Diosmin has been shown to decrease the survival rate of skin cancer cells (A431) compared to the control. The IC50 determined was 45 µg/mL [54].

In many pharmacological studies in animals, diosmin reduced the rolling, adhesion, and migration of leukocytes. In clinical trials, diosmin decreased the expression of monocytes or neutrophils, and the endothelial activation markers of the intercellular adhesion molecule 1 (ICAM-1) and the vascular cell adhesion molecule 1 (VCAM-1) on human leukocytes in patients with venous ulceration [55].

Another drug escin is an anti-inflammatory drug that reduces swelling and improves blood flow. The pharmacological effect of escin is associated with its anti-edema and anti-inflammatory action, effect on the venous tone, and protective effect on the initiation of endothelial damage in hypoxia. In addition, it restores the elasticity and proper tension of blood vessels and increases tissue oxygenation, as well as accelerates the dissolution of blood clots. Due to its positive effect on the vessels, escin is a drug used in the treatment of chronic venous insufficiency including varicose veins. The effect of β-escin sodium up to 40 μg/mL has been shown to inhibit proliferation and migration but also to initiate apoptosis at the maximum concentration in HUVEC and human umbilical vein endothelial (ECV304) cells [56]. β-escin-induced cytotoxicity, assessed at the same time as in our study, was observed at concentrations above 4 μM [57]. This result revealed that the isolated β-escin isomer may show greater toxicity to endothelial cells than an isomer mixture. Escin led to an increase in the antioxidant potential demonstrated with DPPH and FRAP, as well as to the inactivation of O_2_^•−^. A significant decrease in the viability of A2780 cells was also observed after the treatment with escin. However, it did not cause cytotoxicity to the normal Vero cells (immortalized cell line established from the kidney epithelial cells of the African green monkey) [58]. Recent studies confirm the anti-inflammatory properties of escin in decreasing the permeability of blood vessels in inflammation, which leads to the inhibition of edema formation. Moreover, it was found that escin prevents hypoxia-induced disturbances in the proper expression and distribution of platelet endothelial cell adhesion molecule-1(PECAM-1), which may explain its protective effect on vascular permeability [59]. Endothelial dysfunction accompanies the pathogenesis of decompression sickness (DCS), and has a significant influence on the subsequent inflammatory response. It was shown that the administration of escin significantly reduced the mortality of rats caused by DCS. In addition, it alleviated endothelial dysfunction, led to a decrease in serum E-selectin and ICAM-1, and decreased the level of pro-inflammatory cytokines IL-6 and TNF-α in the serum and MPO, as well as the MDA of the oxidative stress marker [60]. It was shown that β-escin had an inhibitory effect on proliferation, migration, and tube formation induced by basic fibroblast growth factor (bFGF) in HUVEC cells, and that it decreased the viability of concentration-dependent cells in vitro. This saponin also inhibited critical steps in the angiogenic process by β-escin, which may be related to the suppression of the Akt (protein kinase B) activation in response to bFGF [61].

The effect of escin has been studied in glioma and lung adenocarcinoma cell lines, C6 and A549 cells, respectively, as well as the ovarian cell line A2780 cells. This saponin has been shown to have a strong antiproliferative effect on C6 and A549 glioma cells. Escin led to cell cycle arrest in the G0/G1 phase and an increase in selective apoptotic activity in A549 cells, which is associated with an increase in annexin V binding, bax protein expression, and caspase-3 activity, but also with observed morphological changes in cells [62]. Our studies showed that the effect of bromelain on the viability of HUVEC and HVVEC was similar at all tested concentrations. Bromelain showed the lowest cytotoxicity compared to diosmin and escin. Bromelain did not influence RAW264.7 cell viability in concentrations of 10–80 µg/mL [63]. Our results showed that endothelial cells were more sensitive to bromelain treatment than RAW264.7 cells. On the other hand, a strong effect of bromelain on the viability of low cells at concentrations up to 25 µg/mL was found in the two tumor cell lines: human oral squamous carcinoma (SCC-25) and Ca9-22, but not in the human keratinocyte HCaT and human gingival fibroblast HGF-1 lines [64]. This results confirm that bromelain is more sensitive to tumor cell lines than to normal cell lines.

### 4.2. Reactive Oxygen and Nitrogen Species Generation

The tested drugs, apart from their antioxidant properties, also showed a pro-oxidative effect. In our studies using the H_2_DCF-DA probe, which reacts with many ROS, we found that escin slightly increased the fluorescence in HUVECs. A slight increase in the fluorescence was also observed in HVVECs following the treatment with bromelain. The generation of ROS using H_2_DCF-DA was also observed in normal and cancer cells following treatment with diosmin, escin, and bromelain. The excessive release of ROS under the influence of diosmin has been shown. ROS were the cause of the apoptosis of A431 cells. Diosmin could inhibit cell proliferation through ROS-mediated apoptosis. The apoptotic properties of diosmin were also tested by performing a DNA fragmentation test [54]. It was shown that an exposure of PMN cells to a mixture containing bromelain (Wobenzym) caused a significant increase in the release of ROS detected by the chemiluminescence method. A significant increase in the ROS generation was also observed after the administration of Wobenzym to healthy volunteers [65]. Moreover, p38 mitogen-activated protein kinases (MAPKs) and reactive oxygen species were activated by escin. Escin induced ROS formation in a dose-dependent manner. A p38 MAPK inhibitor partially attenuated the autophagy and apoptosis triggered by escin, but a ROS scavenger showed a greater inhibitory effect in human osteosarcoma cells in vitro and in vivo [66]. On the other hand, escin led to an increase in the lipid peroxidation and suppressed the SOD activity and glutathione (GSH) level in A2780 cells. A drastic increase in ROS generation, and apoptotic cells, following the saponin treatment was also observed. Escin suppressed the p38 MAPK/ERK signaling axis in A2780 cells [58]. Diosmin induced significant ROS generation in tumor cells: human breast cancer cells MCF-7, MDA-MB-231, and SK-BR-3 treated with diosmin [67]. Similar results were found in the prostate cancer cell DU145 [68].

We also used a DHEt-specific probe to estimate the superoxide generation. We observed a slight increase in the fluorescent intensity in HVVEC after the diosmin and escin treatment, but the differences were statistically insignificant. Superoxide production was found using the DHEt fluorescent probe after the treatment of human breast cancer cells MCF-7, MDA-MB-231, and SK-BR-3 with diosmin [67], and also in prostate cancer DU145 [68]. Escin has been shown to induce apoptosis in human renal cancer cells by producing ROS, which was detected by DHEt [69]. Superoxide was also found in escin-treated human bladder cancer cells, where ROS induced apoptosis [70]. On the other hand, the antioxidant effect of diosmin and bromelain was observed in red blood cells. Both compounds induced an increase in total thiols and glutathione, and a decrease in carbonyls. The increase in GSH concentration and the total level of thiol compounds favored the protection of RBCs against oxidative stress [71].

Another specific probe for nitric oxide measurement, DAF-FM, showed a slight increase in the fluorescence following the diosmin treatment of HUVEC cells. A similar result for this probe was also found in HUVECs after the treatment with bromelain. These results can be indicative of a slight release of ^•^NO in HUVEC cells. However, these results were not statistically significant. The production of nitric oxide in tumor cells has been demonstrated, i.e., in human breast cancer cells MCF-7, MDA-MB-231, and SK-BR-3 treated with diosmin using the MitoSOX™ probe [67]. In vitro and in vivo studies have shown that the administration of bromelain prevents the aggregation of human platelets [72,73]. Fibrinolytic activity has also been demonstrated, as well as the inhibition of thrombus formation [74]. It was shown that the incubation of platelets with bromelain completely prevented thrombin-induced platelet aggregation. In addition, bromelain decreased the adhesion of thrombin-stimulated platelets to bovine aortic endothelial cells in vitro. In a laser-initiated thrombosis model in rats, orally-administered bromelain inhibited thrombus formation. However, when given intravenously, it was somewhat more effective. These studies also showed that orally administered bromelain is biologically active [72]. It was reported that bromelain stimulated the inflammatory mediators, such as interleukin IL-1β, IL-6, interferon (INF-γ), and tumor necrosis factor (TNF-α), in mouse macrophage and in human mononuclear cells [75,76]. Another report showed that bromelain can stimulate a healthy immune system in connection with a fast response to cellular stress. On the other hand, bromelain decreases the release of pro-inflammatory cytokines in activated immune cells when it occurs during their expression [77,78].

A potent anti-inflammatory effect of bromelain has been demonstrated in RAW264.7 cells treated with LPS. A reduction in cytokines and inflammatory mediators has been observed. Bromelain led to a decreased expression of iNOS and COX-2, NF-κB (nuclear factor kappa B), as well as extracellular signal-regulated kinases and the amino-terminal c-Jun kinase and p38 protein in LPS-treated cells [63]. Bromelain inhibited the activity of adhesion proteins on the surface of leukocytes responsible for the adhesion and migration of inflamed cells. Changes in the protein activity were observed at a concentration of 1 µg/mL [79].

### 4.3. Cell Membrane Fluidity

We also determined the fluidity of the plasma membranes of human umbilical vein endothelial cells and endothelial cells derived from a varicose vein using two fluorescent probes located at different depths of the lipid monolayer of the membrane. By using the TMA-DPH probe, we showed a lower fluidity of the HVVEC membrane than of HUVEC. Even greater differences in the membrane fluidity were observed with the second DAUDA probe, located in the deeper region of the monolayer. These results show a higher stiffness of the HVVEC plasma membranes compared to HUVEC cells. The cell membrane is an organized structure (lipid bilayer) composed of phospholipids, proteins, and carbohydrates as a result of lipid–lipid and lipid–protein interactions; together they create a fluid mosaic. The movement of molecules (molecular motion) in the membrane can be monitored with the use of spin probes or fluorescent probes. One of the parameters characterizing the membrane is the fluidity of the membrane, which depends on the chemical structure of phospholipids, including the degree of saturation of fatty acids, as well as the protein–lipid ratio and the presence of cholesterol in the membrane. The plasma membranes have a gradient of fluidity from the water interface with the interior of the bilayer [80,81]. The membrane flexibility is closely related to the transport of the substance across the membranes into and out of the cell, both passive and active [82,83]. The increase in the HVVEC plasma membrane stiffness may be a consequence of the oxidative stress that occurs in a varicose vein. Moreover, the adhesion of leukocytes to the vessel wall may promote additional oxidative stress and the remodeling of the vessel wall. Changes in the structure of VV-derived RBCs may reflect changes that occur in other cells derived from the pathological vein, including endothelial cells.

A significant decrease in the fluidity of the lipid membrane of erythrocytes (RBC) derived from varicose veins, in comparison to the antecubital vein, was also observed using spin-labeled 5-doxylstearic acid (5-DS), which is located in the polar region of the lipid layer. In addition, we demonstrated the changes in the structure of the spectrin–actin complex, which is an important component of the membrane cytoskeleton. We also found a decrease in the internal viscosity of RBCs from varicose veins. These changes were accompanied by the greater osmotic sensitivity of RBCs from a varicose vein [84]. These changes have an influence on the rheological properties of RBCs, which are related to their deformation when they pass through capillaries in the microcirculation. Rheological properties of cells are determined by the viscosity of the interior, the fluidity of plasma membranes, and the condition of the membrane cytoskeleton.

We also found an increased level of oxidative stress in blood derived from a varicose vein compared to blood from an antecubital vein, resulting from a failure of the antioxidant defense system. The observed changes in RBC (VV) properties could be caused by the permanent oxidative stress prevailing in the varicose vein environment. In our previous paper, we compared the antioxidant status of plasma and RBC obtained from peripheral veins and varicose veins in the same patients. We demonstrated a decrease in non-enzymatic antioxidant capacity (NEAC) in plasma obtained from varicose veins compared to peripheral veins. These changes were accompanied by a decrease in the activity of catalase (CAT) and acetylcholinesterase (AChE). We observed a decrease in the level of thiols in plasma, hemolysate, and plasma membranes, and an increase in parameters related to oxidative stress, such as the level of protein carbonyl compounds and the level of thiobarbituric acid reactive substances (TBARS) in varicose veins [85].

Diosmin and bromelain reduced the internal viscosity of erythrocytes and led to a significant decrease in the mobility of the spin label attached to cytosolic thiols. In addition, a change in hemoglobin conformation was observed at higher diosmin concentrations and for both bromelain concentrations. Both compounds tended to reduce cell membrane fluidity in the subsurface area, but not in deeper areas. A higher concentration of bromelain led to a significant decrease in membrane fluidity. In addition, both compounds showed a stabilizing effect on the cell membrane, and improved the rheological properties of RBCs [71].

Many studies have shown that membrane fluidity is crucial for the cell adhesion mechanism [86,87]. Membrane fluidity also has an influence on cell–cell communication [88,89]. Cell adhesion is optimized by building nanoclusters of ordered lipid rafts that are associated with adhesive complexes [90]. Lipid rafts are fluid domains heterogeneously distributed throughout the membrane. A more fluid membrane can also help to disperse the lipid rafts quickly, thus reducing adhesions [86,87]. Using the 5-DS spin probe, it was shown that compared to growing MT3 breast cancer cells, confluent cells had a significantly higher membrane fluidity, which was associated with a higher relative proportion of unordered domains, and a smaller proportion of the most ordered domains. The higher fluidity of the membranes led to a lower adhesion of cells to the surface and a higher invasion of these cells [91]. Both types of cells responded differently to the drug treatment. Diosmin and escin caused an increase in the HVVEC membrane fluidity and a decrease in the HUVEC fluidity in the polar region of the monolayer of the plasma membrane. A similar trend was observed for both types of cells for escin and bromelain in the hydrophobic region membrane. In contrast, diosmin led to a decrease in the membrane fluidity for both HUVEC and HVVEC cells. An in vitro analysis of cellular and molecular responses in human endothelial cells in inflammation showed that β-escin strongly induced cholesterol synthesis, which resulted in a decrease in the integrity of the actin cytoskeleton in endothelial cells. As a consequence, there were changes in cell function which caused a significantly reduced response to the TNF-α stimulation. These included decreased migration, decreased endothelial monolayer permeability, and the inhibition of NF-κB signal transduction, leading to a downward expression of TNF-α-induced effector proteins [57]. This result is consistent with our results because escin led to an increase in the membrane fluidity, which is usually accompanied by an increase in the mobility of the spectrin–actin complex included in the membrane cytoskeleton. In our earlier work, we showed that the decrease in the fluidity of the membrane was correlated with the decrease in the motion of the spin labeled spectrin–actin complex [84]. It was reported that the pre-incubation of platelets with bromelain, prior to thrombin activation, decreased the platelet adhesion to endothelial cells to a value characteristic of unstimulated platelets [72].

The obtained results indicate that the tested drugs at the concentrations of 1 and 10 µg/mL did not induce significant changes in the fluidity of the cell membrane at different depths of the monolayer in both types of endothelial cells. These results are consistent with cytotoxicity and reactive oxygen species studies, which showed no deleterious effects of diosmin, escin, and bromelain on endothelial cells. Our findings suggest that the presented abnormalities in the endothelial plasma membrane may have significant pathophysiological implications, including varicose vein thrombosis. The tendency for the membrane fluidity of HVVECs to increase following treatment with diosmin, escin, and bromelain may lead to a lower adhesion of blood cells to the endothelium, and, consequently, to a lower risk of vein thrombosis.

## 5. Conclusions

We found that a decrease in the fluidity of the endothelial cell membrane in patients with varicose veins may cause increased cell adhesion properties. This may be the reason for a much more frequent occurrence of venous thrombosis in these patients. We demonstrated, for the first time, the mechanism of action of anti-varicose drugs (diosmin, escin, and bromelain) on endothelial cells derived from a varicose vein.

## Figures and Tables

**Figure 1 biomedicines-11-01702-f001:**
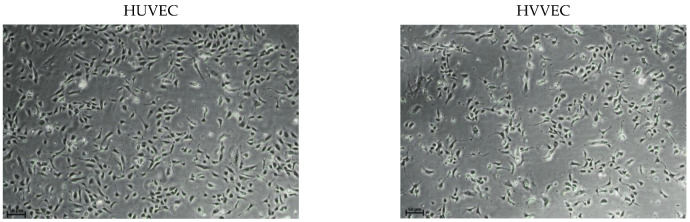
Microscopy images of human umbilical vein endothelial cells (HUVEC) and vascular endothelial cells (HVVEC) cultured in an MCDB131 medium (scale bar 50 μm).

**Figure 2 biomedicines-11-01702-f002:**
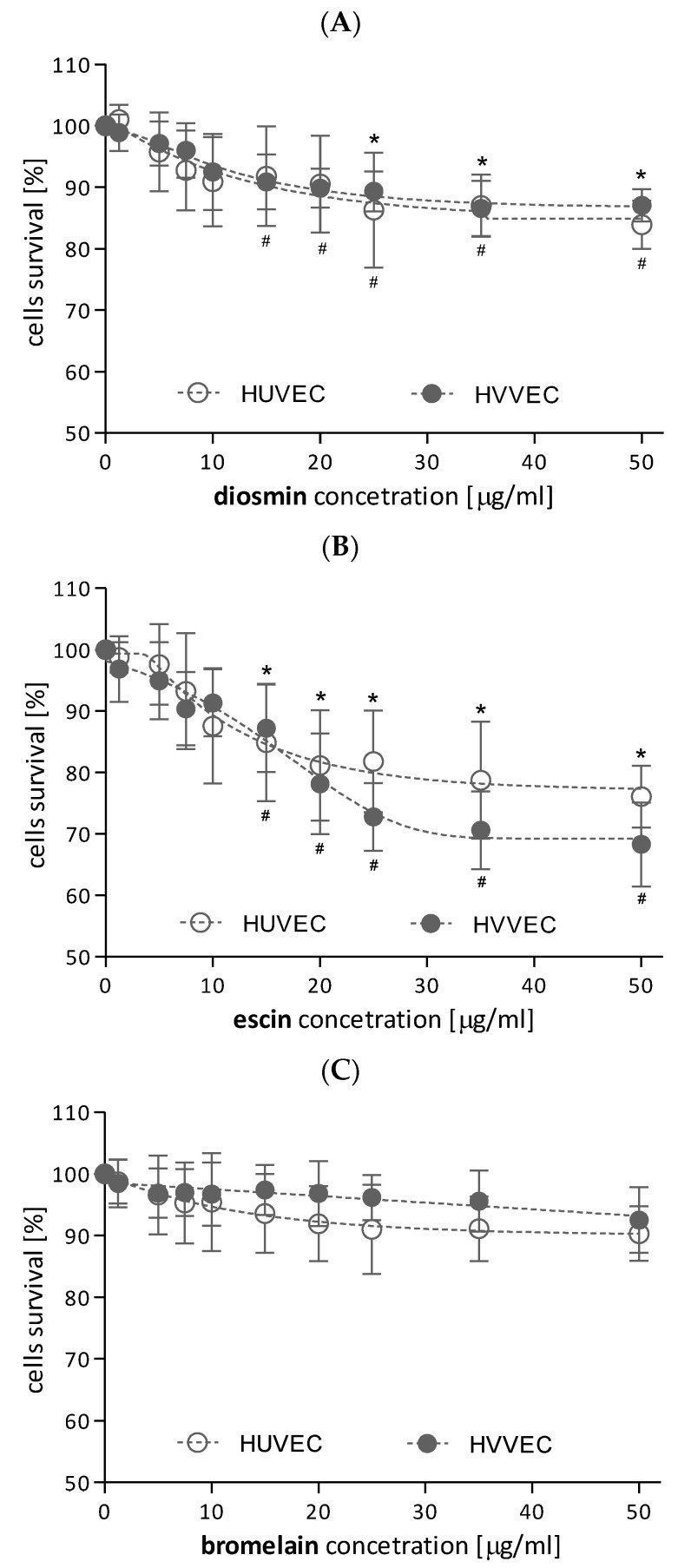
Survival curves of human umbilical vein endothelial cells (HUVEC) and human varicose vein endothelial cells (HVVEC) treated with diosmin (**A**), escin (**B**), and bromelain (**C**). The results represent mean ± SD of data from six individual experiments for varicose and umbilical veins, each conducted with at least eight repeats. * *p* < 0.05 versus control of HUVEC, and ^#^ *p* < 0.05 versus control of HVVEC.

**Figure 3 biomedicines-11-01702-f003:**
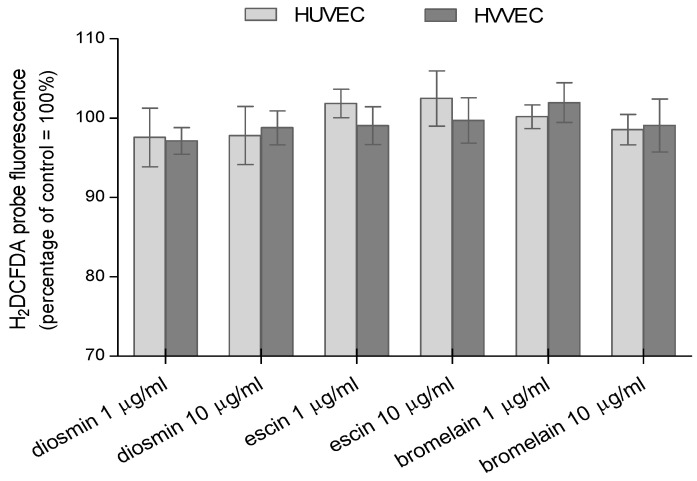
Relative amounts of reactive oxygen species generated in human umbilical vein endothelial cells (HUVEC) and human varicose vein endothelial cells (HVVEC) treated with diosmin, escin, and bromelain. The results represent mean ± SD of data from six individual experiments for varicose veins and for umbilical veins, each conducted with at least eight repeats.

**Figure 4 biomedicines-11-01702-f004:**
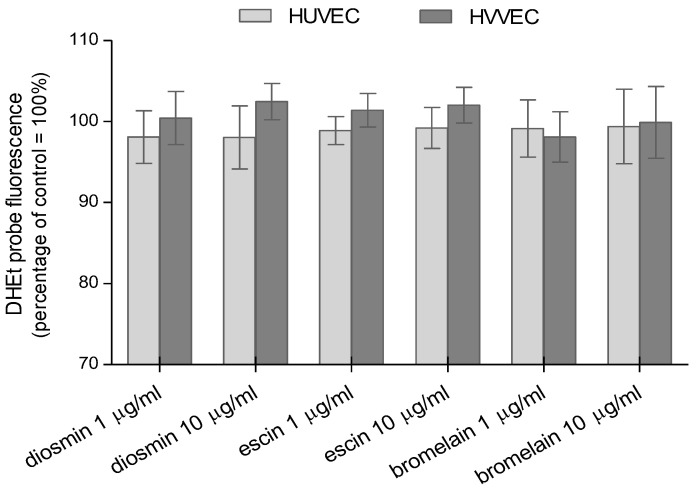
Relative amounts of superoxide anion radical generated in human umbilical vein endothelial cells (HUVEC) and human varicose vein endothelial cells (HVVEC) treated with diosmin, escin, and bromelain. The results represent mean ± SD of data from six individual experiments for varicose veins and for umbilical veins, each conducted with at least eight repeats.

**Figure 5 biomedicines-11-01702-f005:**
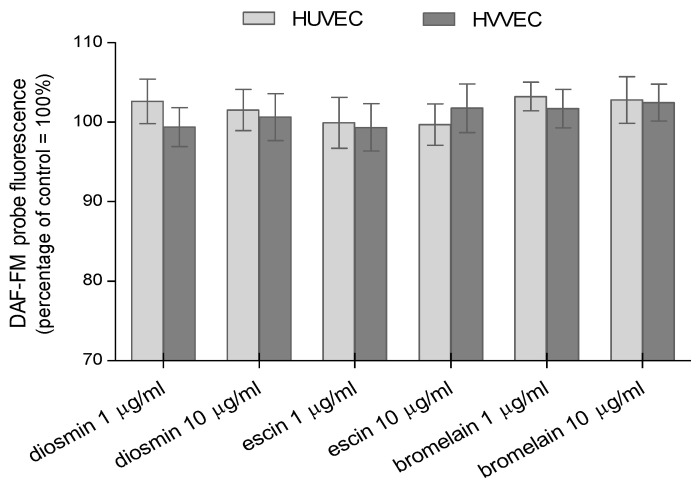
Relative amounts of nitric oxide generated in human umbilical vein endothelial cells (HUVEC) and human varicose vein endothelial cells (HVVEC) treated with diosmin, escin, and bromelain. The results represent mean ± SD of data from six individual experiments for varicose veins and for umbilical veins, each conducted with at least eight repeats.

**Figure 6 biomedicines-11-01702-f006:**
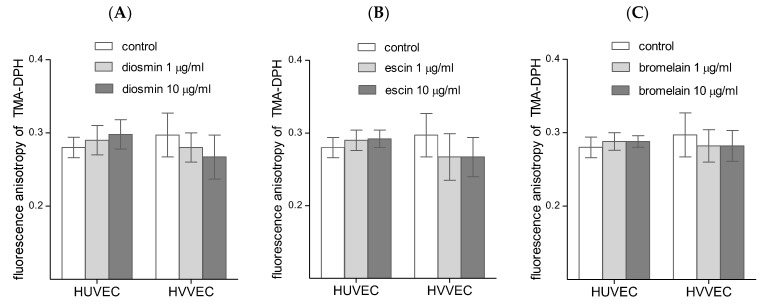
Fluorescence anisotropy of a TMA-DPH label in human umbilical vein endothelial cells (HUVEC) and vascular endothelial cells (HVVEC) treated with diosmin (**A**), escin (**B**), and bromelain (**C**). The results represent mean ± SD of data from six individual experiments for varicose veins and for umbilical veins, each conducted with at least six repeats.

**Figure 7 biomedicines-11-01702-f007:**
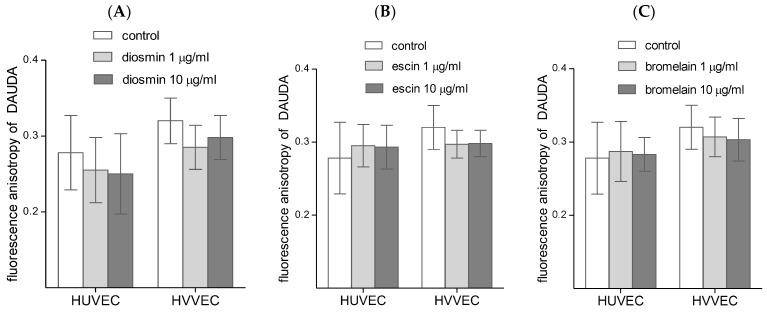
Fluorescence anisotropy of a DAUDA label in human umbilical vein endothelial cells (HUVEC) and vascular endothelial cells (HVVEC) treated with diosmin (**A**), escin (**B**), and bromelain (**C**). The results represent mean ± SD of data from six individual experiments for varicose veins and for umbilical veins, each conducted with at least six repeats.

## Data Availability

The data presented in this study are available on request from the corresponding authors.

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
