# Peer review of "The Effect of Diosmin, Escin, and Bromelain on Human Endothelial Cells Derived from the Umbilical Vein and the Varicose Vein—A Preliminary Study"

_biomedicines, 2023, doi:10.3390/biomedicines11061702_

Round 1

Reviewer 1 Report

The manuscript submitted to the Biomedicines by Dr. Lukasz Gwozdzinski et al. and entitled «The effect of diosmin, escin, and bromelain on human endothelial cells derived from the umbilical vein and the varicose vein – preliminary study» is aimed to in vitro study of drugs in the therapy of varicose vein. Authors showed the mechanism of action of drugs used in varicose vein therapy on endothe-lial cells derived from a varicose vein. The study is well performed and the results are clearly presented. I have only two minor issues that must be solved before publication.

1. Why HUVEC was selected for the comparative analysis?

2. Please add the ruler on the all images included in the Figure 1.

Author Response

Thank you very much for sending us Your comments on our paper entitled: “Indoxyl sulfate induces oxidative changes in plasma and hemolysate”

Most of the answers to questions and suggestions were placed in a revised version of the manuscript. Below, we enclose the responses.

We would like to add that the spelling and grammatical mistakes in the manuscript have been checked and are now corrected by a professional.

Thank you for the opportunity to correct the manuscript.

Reviewer 2 Report

Very important study on the molecular benefits of diosmin, escin, and bromelain. Following are my few suggestions to improve the manuscript.

Abstract:

- explain the meaning of ROS, H2DCFDA, DHEt, DAF-FM, TMA-DPH and DAUDA.

Introduction:

- if you introduce an acronym (i.e. VV, MMP, CVD), use it in the following text;

- “…and are the symptom of chronic venous insufficiency”: better to say “…and are the signs of chronic venous insufficiency”;

- explain the meaning of SOD when first use it, and VEGF.

Materials and Methods:

- explain the meaning of DMSO, PBS, BMI, RNS;

- if you introduce an acronym (i.e. CVD, DHEt), use it in the following text;

- no need to repeat the meaning of an acronym used in the text before (i.e. H2DCFDA, TMA-DPH, DAUDA).

Results:

- explain the meaning of DCFH-DA;

- no need to repeat the meaning of an acronym used in the text before (i.e. O2•-).

Discussion:

- VV is better than “Vein morphology…”

- if you introduce an acronym (i.e. VV, ICAM, VCAM, O2.-), use it in the following text;

- explain the meaning of IC50, ERK;

- “The observed changes in RBC (VV) properties…”: what VV does it mean here? You have already used VV for varicose veins.

Since there are a lot of acronyms, I suggest to introduce a “Glossary” of all the acronyms at the beginning of the manuscript.

Minor English revision required.

Author Response

(The authors gave the same response as above.)
